# Integrative multi-omics and network-based machine learning for early diagnosis of Parkinson's disease

Wei Liu[1], Lina Xu[2], Xuejing Wang[1]*, Jiuqi Wang[1]*

1 Department of Neurology, The First Affiliated Hospital of Zhengzhou University, Zhengzhou, China,
2 Department of Neurology, Jincheng people's hospital, Jincheng, China

* fccwangxj2@zzu.edu.cn (XW); jiuqi970224@163.com (JW)

## Abstract

### Background

Accurate diagnosis of Parkinson's Disease (PD) remains challenging due to its biological complexity. Integrating machine learning with multi-omics and network topological analyses may enhance diagnostic precision.

### Objective

To classify early-stage PD patients and healthy controls (HCs) by integrating multi-omics data and network-based machine learning.

### Methods

We analyzed 305 participants from the PPMI cohort (213 PD, 92 HCs) using DNA methylation, gene expression, and proteomic data. Feature selection was conducted through sPLS-DA, followed by integration via DIABLO. Regulatory gene networks were constructed using STRING-based topology analysis. An XGBoost classifiers were trained and optimized on 244 samples and validated on 61 independent samples. Furthermore, we conducted external validation using data from 26 participants (12 PD, 14 HCs) in the GEO dataset, incorporating DNA methylation and gene expression profiles.

### Results

DIABLO-based integration identified 56 CpG sites, 61 genes, and 70 proteins. Network topology analysis revealed 59 key regulators. Among three XGBoost models—based on multi-omics signatures, topological regulators, and their combination—the multi-omics model achieved the best test-set performance (AUC 0.72, 95%CI 0.51-0.85, accuracy 0.74, 95%CI 0.61-0.82). Within the validation set, the topological regulators model achieved the superior performance (AUC 0.57, 95%CI 0.37-0.73, accuracy 0.62, 95%CI 0.38-0.73).

**Data availability statement:** All relevant data used in this study are publicly available from two established third-party repositories. The Parkinson's Progression Markers Initiative (PPMI) [https://www.ppmi-info.org/access-data], supported by the Michael J. Fox Foundation, is available with open access to all qualified researchers via registration and acceptance of a standard Data Use Agreement. The Gene Expression Omnibus (GEO) [https://www.ncbi.nlm.nih.gov/geo/query/acc.cgi?acc=GSE165083] database is publicly available under accession number GSE165083.

**Funding:** The author(s) received no specific funding for this work.

**Competing interests:** The authors have declared that no competing interests exist.

## Conclusion

Combining machine learning with integrative multi-omics and network topology analysis enables effective biomarker identification and PD classification, with strong potential for clinical diagnostic applications.

---

## Introduction

Parkinson's disease (PD) stands as the second most common neurodegenerative disorder among the elderly, affecting up to 4% of individuals by the age of 80 [1]. It is primarily characterized by motor symptoms such as bradykinesia and rigidity, along with non-motor features including cognitive and autonomic dysfunction, which together impair quality of life [2,3]. Although extensive research has elucidated several molecular and pathological features of PD, it remains an incurable condition; current therapeutic approaches are limited to symptomatic relief and do not modify disease progression [4]. A major clinical challenge lies in the typically late-stage diagnosis of PD, which often occurs years after the onset of neurodegeneration. By the time motor symptoms emerge, a substantial proportion of dopaminergic neurons has already been lost, greatly limiting the window for effective therapeutic intervention [5]. Therefore, the identification of sensitive and specific biomarkers for early diagnosis represents a significant unmet need in the management of PD [6–8].

Given the heterogeneity and multifactorial nature of PD, which encompasses a complex interplay of genetic, epigenetic, metabolic, and inflammatory processes, the identification of robust and clinically useful biomarkers will likely depend on the integration of multiple molecular indicators [9,10]. In this context, high-throughput "omics" technologies—including genomics, transcriptomics, proteomics, and metabolomics—have emerged as powerful tools for unbiased, system-level biomarker discovery by enabling comprehensive profiling of molecular changes across disease stages [11,12]. As a result, omics-driven approaches have increasingly contributed to the discovery of candidate biomarkers in PD and other neurodegenerative disorders, paving the way for earlier diagnosis, patient stratification, and precision medicine applications [13,14].

Machine learning (ML) has emerged as a powerful tool in the integration and analysis of high-dimensional biomedical data, offering promising applications for disease classification and biomarker discovery in PD [15,16]. By incorporating diverse inputs—including neuroimaging features, genetic risk variants from genome-wide association studies (GWAS), and transcriptomic profiles, ML approaches have demonstrated improved accuracy in PD diagnosis and subtype prediction [17–20]. Beyond classification, advanced network-based algorithms such as Hidden Nodes and Network Propagation enable the identification of upstream regulatory genes based on their topological importance within molecular interaction networks [21,22]. These integrative strategies not only enhance diagnostic precision but also provide mechanistic insights into the molecular drivers of disease.

In our study, we employed machine learning techniques to integrate multiple sources of biological information, encompassing DNA methylation arrays, gene expression data, and proteomic profiles. This integrative methodology facilitated the classification of early PD patients from healthy controls (HCs), showcasing potential applications in clinical diagnostics. To enhance target identification, we integrated network algorithm pipelines utilizing the protein interaction database STRING. Furthermore, our analysis identified novel candidate biomarkers and illuminated the interconnections among these markers across various omics layers.

## Materials and methods

### PPMI and GEO dataset: data preprocessing

DNA methylation arrays, gene expression profiles, and proteomic data were obtained from the Parkinson's Progression Markers Initiative (PPMI; www.ppmi-info.org/data) in October 2024. Study design and sample collection protocols have been described in detail in previous publications [23]. Patients diagnosed with early PD had not been treated with dopaminergic medications and did not carry mutations in *LRRK2*, *GBA*, or *SNCA* genes. Additionally, individuals with first-degree relatives carrying any of these mutations were excluded. Age- and sex-matched HCs were neurologically normal and free of medications affecting the dopaminergic system, including dopamine receptor antagonists [24].

All samples were divided into training (80%) and testing (20%) cohorts using stratified random sampling. Single-omics and multi-omics bioinformatic analyses, including functional enrichment and network topology inference, were performed on the training set. ML was conducted using the full dataset; however, the model was trained using only the training data. Notably, the testing dataset was reserved solely for evaluating the performance of the model, which served as internal validation. To further assess generalizability, an external validation dataset (GSE165083) was downloaded from the Gene Expression Omnibus (GEO) database (https://www.ncbi.nlm.nih.gov/geo/) (Fig 1). This validation set comprised DNA methylation and expression profiles of whole blood samples from PD patients and matched controls [25], which was utilized to verify the performance of the ML model.

Since not all samples had complete multi-omics data, we included only those with complete measurements across all omics layers to minimize potential bias in downstream analyses, particularly in latent component estimation.

**DNA methylation.** In the PPMI cohort, the DNA methylation array data from whole-blood were normalized using the Functional Normalization method [26], a form of between-array normalization that eliminates undesired variation by regressing out the variability attributable to control probes incorporated within the array. All CpG sites' methylation levels were coded in the form of M-values [27]. To refine the analysis, predictor variables exhibiting low variance across the samples were excluded; consequently, the top 10,000 most variable genes were selected for further investigation. In the GEO database, the methylation analysis was conducted using Illumina Infinium HumanMethylation450K Beadchips. Subsequently, the data were normalized utilizing the BMIQ Method, and any batch effects were appropriately corrected using ComBat [25].

**Gene expression.** Gene expression data from whole-blood were processed using DESeq2 normalization to eliminate technical biases associated with discrepancies in library sizes [28]. Genes that were either not expressed or expressed at low levels, defined as having an unnormalized median read count of less than 1 across all samples, were excluded from the analysis. These were considered likely to be non-informative features. In this dataset, predictor variables exhibiting low variance across all samples were excluded. Consequently, we focused on the top 5000 genes with the highest variability for subsequent analysis. In the GEO dataset, the gene expression analysis was consistent with the findings from the PPMI cohort.

**Proteomics.** Proteomic data from cerebrospinal fluid were analyzed using the SOMAscan platform. To ensure quality, outlier samples, calibrators, buffers, and non-human SOMAmers were excluded. The obtained measurements underwent hybridization normalization, plate scaling, and median normalization both intra-plates. Subsequently, these values were

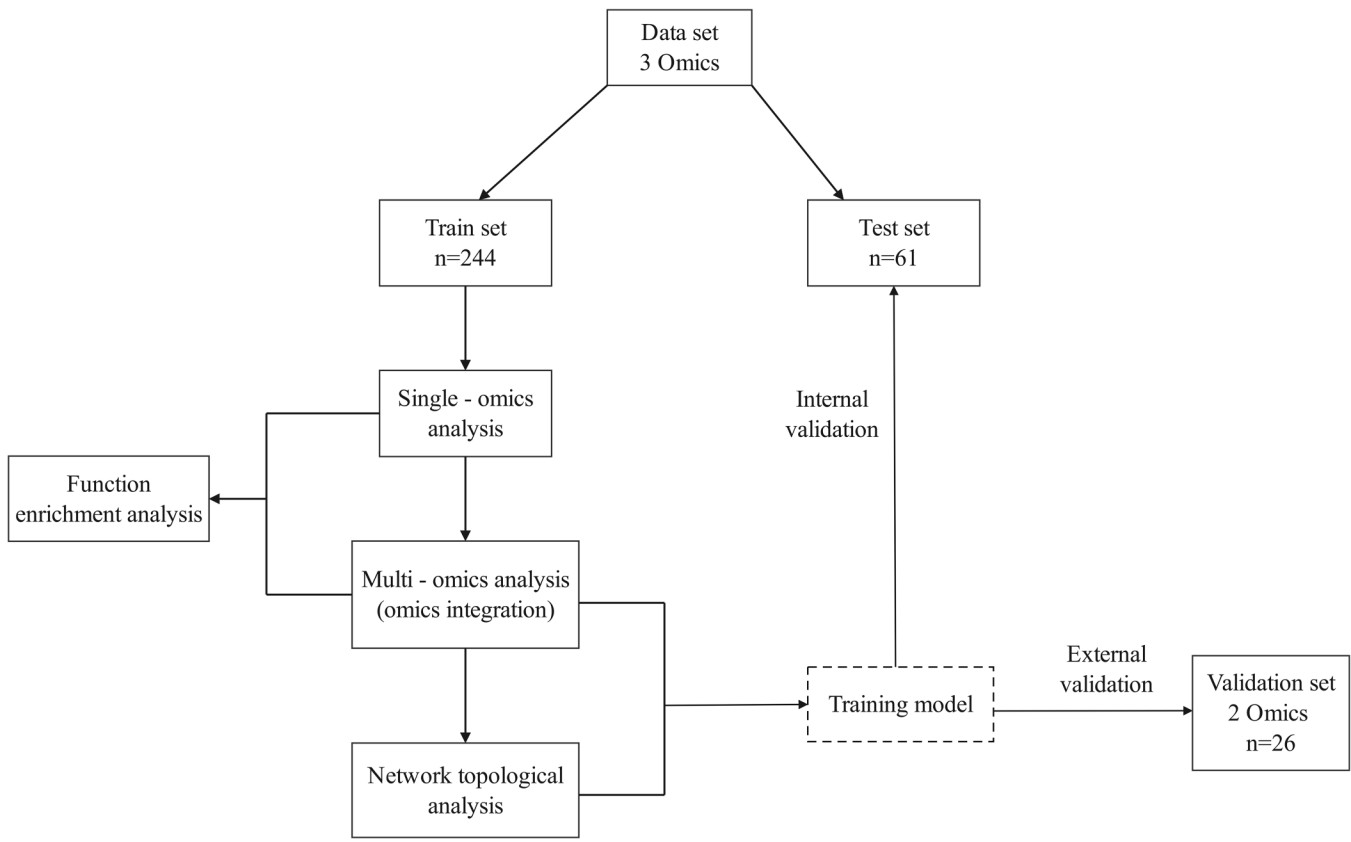

**Fig 1. A schematic presentation of our study.** The full dataset is randomly split into train (n = 244, 80%) and test (n = 61, 20%) datasets. Single-omics and multi-omics bioinformatic analyses, including functional enrichment and network topology inference, were performed in the training dataset. Machine learning was conducted using the full dataset, the model was developed using training data and its performance was evaluated using testing data. Additionally, the model's performance was externally validated using an independent dataset.

calibrated at SomaLogic and transformed using a $\log_2$ function. They were then median normalized between plates and underwent batch correction at the plate level.

## Single- and multi-omics data preprocessing and integration analysis

Single-omics analyses were performed on DNA methylation, gene expression, and proteomic datasets using sparse partial least squares discriminant analysis (sPLS - DA) [29,30]. This multivariate method enables sample classification and feature selection by projecting the data into a lower dimensional space in a supervised manner. Given the inherent characteristics of omics data—namely, large feature sets and limited sample sizes—sPLS-DA incorporates feature pre-selection to mitigate the curse of dimensionality. For multi-omics analysis on multiblock datasets, which were derived from three omics data, we utilized integrative sPLS-DA via the Data Integration Analysis for Biomarker discovery using Latent cOmponents (DIABLO) algorithm [31] provided in mixOmics R package (http://mixomics.org/, version 6.12.1), operating in the R 4.4.2 environment. DIABLO extends sPLS-DA to multiblock datasets by identifying correlated latent components across omics layers, thereby enabling joint feature selection and classification. This approach maximizes covariance between omics-derived components and facilitates the discovery of coordinated molecular signatures underlying complex biological phenotypes [32].

## Parameter optimization

Parameter optimization for sPLS-DA involved tuning two key elements: the number of components and the number of features per component. The component number was systematically evaluated from 1 to 6, based on empirical evidence that performance typically declines beyond five components. Feature selection was then optimized by exploring varying feature counts per component. For single-omics models (DNA methylation, gene expression, proteomics), the number of features ranged from 3 to 300, with incremental steps of 3 (3–30), 6 (30–60), 15 (60–150), and 30 (150–300). For integrative sPLS-DA (DIABLO), to ensure consistency across omics layers, feature numbers were uniformly varied from 10 to 50 in steps of 5, yielding nine candidate values per component per dataset. Additionally, a design matrix specifying inter-omics correlations was required. We defined the matrix with a correlation coefficient of 0.1 between omics layers, reflecting moderate biological connectivity.

   Optimal parameters were selected based on model performance under stratified five-fold cross-validation with 100 random permutations. To account for class imbalance and ensure unbiased assessment, model performance was evaluated using the balanced error rate (BER). Final model validation was conducted using stratified five-fold cross-validation repeated 1,000 times, with predictive performance assessed via classification accuracy and receiver operating characteristic (ROC) curve analysis. Furthermore, to quantify the reproducibility of feature selection in sPLS-DA, we conducted a bootstrap stability analysis with 1,000 iterations (B = 1,000), computing selection frequencies for each feature across resampled datasets.

## Functional enrichment analysis

Functional enrichment analysis was performed using the clusterProfiler R package (version 3.12.0) to identify the biological functions of the selected gene sets. The analysis queried three categories of Gene Ontology (GO) datasets: Biological Process (GOBP), Molecular Function (GOMF), and Cellular Component (GOCC), along with pathways from the Kyoto Encyclopedia of Genes and Genomes (KEGG). To control for false discovery rate, Benjamini–Hochberg adjusted $p$-values were calculated. Only terms with adjusted $p$-values below a specified threshold were considered significant. Comprehensive results from both single and integrative sPLS-DA are provided in Table S2 in S1 File.

## Network topological analysis

Protein–protein interaction networks were constructed using the Search Tool for the Retrieval of Interacting Genes/Proteins (STRING; https://string-db.org/) database, which integrates evidence from seven analytical sources, including experimental data, computational predictions, co-expression, text mining, and evolutionary conservation [33]. This resource provided a comprehensive, confidence-weighted framework to map functional interactions among genes and proteins within the multi-omics signatures.

   Topological properties provide insights into the arrangement of components (nodes and edges) within a network and its relevant substructures. To systematically identify topologically significant genes associated with newly identified multi-omics signature, we employed two complementary network-based analytical approaches. First, we employed an algorithm based on the Hidden Nodes (HN) method [21], which prioritizes nodes that provide high connectivity between seed nodes within a directed network. This method identifies nodes that serve as crucial connectors or bridges between the input features. We evaluated the statistical significance of the overconnected nodes using a hypergeometric test and corrected the $p$-values using all internal nodes. The scores correspond to the resulting corrected $p$-values in $-\log_{10}$ scale. Nodes with corrected $p$ values< 0.05 were filtered out. Second, a network propagation (NP) algorithm [22] was employed, a global method that leverages the entire network topology to identify nodes highly connected to the input nodes. This approach differs from local methods by considering the broader context of the network rather than focusing solely on immediate neighbors. The scoring of nodes is performed iteratively, where the flow is repeatedly distributed from seed nodes to

their neighbors and then further propagated throughout the network. For each node, *p*-values are computed for their scores and corrected for multiple testing. The final scores are given by transforming the corrected *p*-values to a $-\log_{10}$ scale, and nodes with scores in the top 1% were retained.In this study, we conduct a comprehensive evaluation of network stability, examining four interconnected pillars: Network robustness--the ability to maintain global connectivity under perturbation-was evaluated by simulating random node removal (5–50% of nodes in 5% increments, 20 simulations per step), with robustness defined as the ratio of the average size of the largest connected component post-removal to its original size. Node criticality was quantified based on the relative decrease in global efficiency (inverse average shortest path length) upon individual node deletion, identifying hubs essential for network integrity. Structural Stability is characterized by key topological metrics: average path length (information propagation efficiency), clustering coefficient (local modularity), modularity (strength of community structure, with $Q > 0.4$ indicating strong modularity), and degree/betweenness centralization (highlighting the concentration of connections and mediation influence, revealing hierarchical organization and small-world properties). Finally, Regulatory Node Stability specifically examines the topological role of biological regulators, we computed the ratio of average degree between regulatory and non-regulatory nodes (regulator degree ratio) and their average betweenness centrality; a ratio >1 indicates that regulators are more highly connected and centrally positioned within the network, with underscoring their potential role as central hubs and bridges coordinating network activity.

### Machine learning and model validation

We employed the eXtreme Gradient Boosting (XGBoost) algorithm, a tree-based ensemble learning method that iteratively improves prediction accuracy by minimizing residual errors through gradient boosting [34]. In this framework, each weak learner—typically a decision tree—is trained to correct the errors of its predecessor, and final predictions are obtained by aggregating outputs from all learners. Multi-omics data were split into training and testing sets, while data from the GEO database served as an external validation set. Model development and hyperparameter tuning were performed exclusively on the training data using five-fold cross-validation with grid search (GridSearchCV). Optimal parameters were selected based on cross-validation performance. The final model was evaluated on the independent test set (internal validation) and further validated on the external dataset. Prior to modeling, all features underwent z-score normalization to standardize the scale of the input variables.

Model performance was assessed using the area under the receiver operating characteristic curve (AUC) to quantify discriminative ability. Additionally, the F1 score was calculated to evaluate the balance between precision and recall, providing a robust measure of classification performance across datasets.

## Results

### Study Cohort and data refinement

Following the exclusion of samples with incomplete data across all omics layers, a total of 305 participants were included from the PPMI cohort. This comprised 213 patients with PD and 92 HCs. Participants were randomly divided into a training data set (n = 244, including 171 PD and 73 HCs) and test data set (n = 61, including 42 PD and 19 HCs). In the training set, the 171 PD patients had a mean age of 62.37 ± 10.00 years, 38.01% were females, with a mean disease duration of 0.51 ± 0.48 years and an average Hoehn & Yahr (H&Y) stage of 1.58 ± 0.50. The 73 HCs in this set had a mean age of 61.71 ± 11.07 years, with 32.88% females. In the test set, the 42 PD patients had a mean age of 62.60 ± 8.99 years, 35.71% were females, with a mean disease duration of 0.48 ± 0.32 years and an H&Y stage of 1.50 ± 0.51. The 19 HCs in this set had a mean age of 62.08 ± 13.52 years, with 15.79% females. Detailed demographic and clinical characteristics of the participants are summarized in Table 1.

Following data preprocessing, the initial dataset included 35,297 genes, 864,067 CpG methylation sites, and 4,785 proteins. To enhance analytical efficiency and interpretability, we refined the dataset by selecting 10,000 CpG sites and 5,000 genes based on predefined criteria, while retaining all proteomic features for downstream analysis.

**Table 1. Baseline Characteristics of the Enrolled Participants in training and testing datadata.**

| Character | Train data | | Test data | |
|---|---|---|---|---|
| | PD(n = 171) | HCs(n = 73) | PD(n = 42) | HCs(n = 19) |
| Age, y | 62.37 ± 10.00 | 61.71 ± 11.07 | 62.60 ± 8.99 | 62.08 ± 13.52 |
| Sex female | 65(38.01%) | 24(32.88%) | 15(35.71%) | 3(15.79%) |
| Duration from the diagnosis, y | 0.51 ± 0.48 | – | 0.48 ± 0.32 | – |
| off H&Y stage | 1.58 ± 0.50 | – | 1.50 ± 0.51 | – |
| MDS-UPDRS part 1 score | 5.52 ± 4.21 | 3.67 ± 2.77 | 5.47 ± 3.70 | 3.16 ± 2.52 |
| Missing* n | 1 | 1 | 0 | 0 |
| MDS-UPDRS part 2 score | 5.76 ± 3.95 | 0.58 ± 1.14 | 6.64 ± 4.95 | 0.95 ± 1.39 |
| Missing n | 1 | 1 | 0 | 0 |
| MDS-UPDRS part 3 off score | 21.62 ± 9.18 | 1.63 ± 2.67 | 19.21 ± 7.24 | 1.74 ± 2.13 |
| Missing n | 0 | 2 | 0 | 0 |
| MDS-UPDRS total off score | 32.88 ± 13.13 | 5.75 ± 4.47 | 31.60 ± 12.71 | 5.84 ± 4.73 |
| Missing n | 1 | 2 | 0 | 0 |
| UPSIT score | 22.43 ± 8.59 | 34.60 ± 4.83 | 22.81 ± 7.71 | 33.10 ± 6.26 |
| MoCA total score | 27.01 ± 2.38 | 28.27 ± 1.13 | 27.45 ± 2.61 | 28.11 ± 2.77 |
| ESS total score | 5.10 ± 3.43 | 5.44 ± 3.58 | 7.36 ± 3.80 | 5.11 ± 2.77 |
| Missing n | 1 | 1 | 0 | 0 |
| RBDSQ total score | 3.96 ± 2.80 | 3.16 ± 2.32 | 4.57 ± 2.70 | 2.26 ± 1.85 |
| Missing n | 1 | 1 | 0 | 0 |
| SCOPA-AUT total score | 9.08 ± 6.04 | 6.05 ± 3.25 | 11.10 ± 7.50 | 4.79 ± 2.74 |
| Missing n | 4 | 0 | 0 | 0 |

ESS, Epworth Sleepiness Scale; H&Y, Hoehn and Yahr; HC, healthy control; MDS-UPDRS, Movement Disorder Society–sponsored revision of the Unified Parkinson's Disease Rating Scale; MoCA, Montreal Cognitive Assessment; nongenetic; PD, Parkinson's disease; RBDSQ, REM Sleep Behavior Disorder Screening Questionnaire; SCOPA-AUT, Scales for Outcomes in Parkinson's Disease-Autonomic; UPSIT, University of Pennsylvania Smell Identification Test. Data are expressed as mean ± SD or n (%).

*The missing data are discarded directly.

In the external validation dataset, which comprised of 12 PD patients and 14 HCs. The average age of PD patients was 72.91 ± 1.74 years, 33.33% were females. The HCs patients had a mean age of 68.17 ± 1.00 years, 57.14% were females. After data preprocessing, the validation dataset included 485,512 CpG sites and 63,677 genes.

## Single-omics Classification with sPLS-DA

sPLS-DA models were independently constructed for DNA methylation, gene expression, and proteomic datasets. The optimal parameters for model construction are summarized in Table 2 and illustrated in S1 Fig in S1 File. In terms of feature selection reproducibility, all chosen features from the DNA methylation data achieved a stability score exceeding 0.8, demonstrating robust selection. For the gene expression dataset, 86.42% of the selected features surpassed a stability score of 0.8, with 99.48% exceeding a threshold of 0.6. In the protein data, 57.62% of the chosen features achieved a stability score above 0.8, while 98.30% surpassed a threshold of 0.6. Detail data were summarized in S1 Table in S1 File. Specifically, 2 components were retained in DNA methylation and gene expression data, while 4 components were left in proteomic data. These included 57 features from DNA methylation dataset, 383 features from gene expression dataset, and 177 features from proteomic dataset. Detailed characteristics of the datasets used in our analysis are provided in S2 Table in S1 File. As shown in Fig 2A, all three models based on single-omics effectively discriminated PD patients from HCs in the first 2 components, albeit with varying performance levels. In component 2, DNA methylation and protein signatures exhibited a certain differential expression between PD and HCs. However, gene signatures did not show a

**Table 2. The optimal parameters for model building.**

| Component | Single sPLS-DA | | | Integrative sPLS-DA | | |
|---|---|---|---|---|---|---|
| | DNA methylation | Gene expression | Proteomics | DNA methylation | Gene expression | Proteomics |
| Component 1 | 8 | 290 | 11 | 50 | 15 | 35 |
| Component 2 | 50 | 95 | 32 | 15 | 20 | 10 |
| Component 3 | NA | NA | 26 | 50 | 10 | 10 |
| Component 4 | NA | NA | 110 | 10 | 35 | 20 |
| Total unique set | 57 | 383 | 177 | 56 | 61 | 70 |

sPLS-DA, sparse partial least squares discriminant analysis.

difference when compared to component 1. This supports notion that component 2 has some discriminative power than component 1 in DNA methylation profiling and proteomic data analysis. The performance results of three models based on single-omics data under optimal parameter settings in the training dataset revealed that proteomic data achieved the highest classification accuracy, followed by gene expression data, while DNA methylation data exhibited the lowest accuracy. Accuracy values across those three models ranged from 0.6 to 0.8, with AUCs varying between 0.55 and 0.75 (S2A Fig in S1 File).

The distribution of selected features was examined using Pearson correlation coefficients between features and latent components (Fig 2B). Component 1 exhibited strong internal correlations among features, while component 2 displayed a more diffuse pattern, suggesting that component 1 may reflect more coordinated biological pathways. KEGG and GO enrichment analyses revealed significantly associated biological processes (FDR < 0.05, Fig 2C and S2 Table in S1 File). Immune and inflammatory responses were universally implicated across all components, with prominent terms including chemotaxis, leukocyte chemotaxis, granulocyte migration. Cell adhesion and migration as well as plasma membrane and cell surface-associated functions were enriched in all components except component 3, exemplified by pathways such as cell adhesion molecules, integrin binding, myeloid leukocyte migration, external side of plasma membrane, and secretory granule membrane. Ribosome-related functions, particularly ribosomal subunits and cytosolic ribosome biogenesis, emerged as a distinctive feature of Component 1. Additional component-specific enrichments were observed: ubiquitin-mediated protein regulation (Component 2), hypoxia and growth factor signaling (Component 3), and neurodevelopmental and amyloid processing pathways (Component 4). Notably, shared molecular features across multiple components encompassed cytokine receptor activity, vesicle trafficking, and G protein-coupled receptor (GPCR) signaling.

## Integrative multi-omics modeling via DIABLO

Using DIABLO-based integrative sPLS-DA, we constructed a multi-omics classification model incorporating 56 CpG sites, 61 genes, and 70 proteins, selected based on optimized parameters (Table 2). This integrated model achieved comparable or superior performance relative to single-omics models in terms of discrimination, accuracy and AUC (Fig 3A, S2A Fig in S1 File). Feature overlap across components is detailed in S2B Fig in S1 File.

To explore the biological relevance of the multi-omics predictive model, we examined the spatial distribution and interrelationships of selected features across omics layers. The Circular visualization of DIABLO-selected features (Fig 3B) revealed sparse co-localization across DNA methylation, gene expression, and proteomics layers, indicating limited cluster formation across these integrated omics dimensions and underscoring their weak mutual correlation. This suggests that, despite integrative modeling, the molecular features retained distinct layer-specific characteristics. To further quantify these relationships, a Circos plot was generated (Fig 4A), which showed moderate positive correlations between gene expression and proteomic features, while DNA methylation sites exhibited minimal correlation with either of the other two omics layers, further supporting their limited integrative contribution.

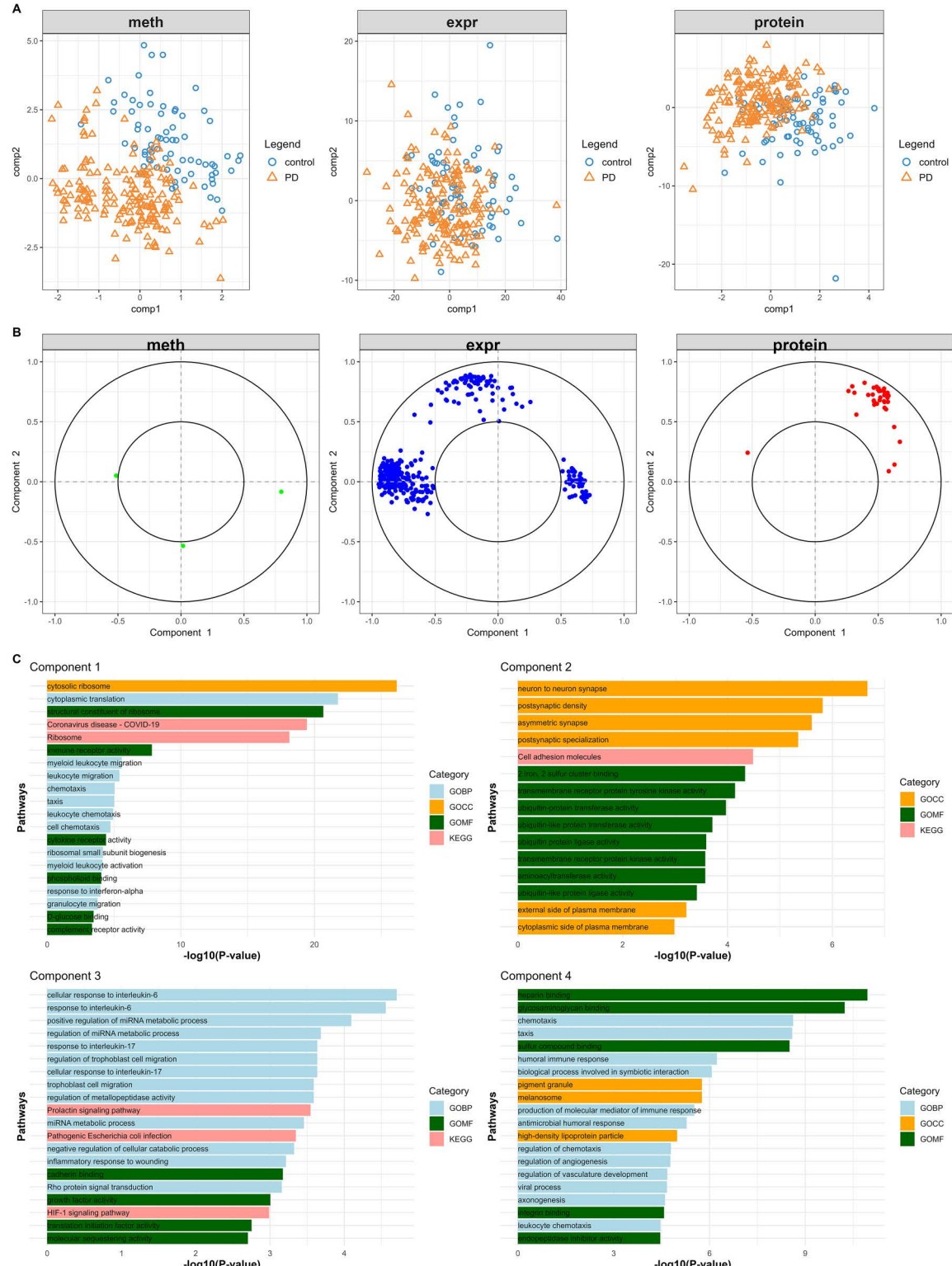

**Fig 2. Single-omics sPLS-DA applied to DNA methylation, transcriptomics, and proteomics data. (A)** Two-component sample distribution (171 PD and 73 HCs) for single-omics sPLS-DA on DNA methylation, gene expression and proteinic data, respectively. **(B)** The selected features from single sPLS-DA across the three omics datasets are plotted according to their Pearson's correlation coefficients with respect to components 1 and 2

(cutoff > 0.5). (**C**)Functional enrichment results on gene set of single sPLS-DA. For each component, 20 highly ranked overrepresented terms are selectively shown. Abbreviations: meth = methylation; expr = expression.

Further investigation of the first latent component—comprising the most abundant feature set—demonstrated robust discriminatory power between PD and HC samples. Heatmap visualization (Fig 4B) revealed distinct clustering aligned with disease status, not driven by any single omics type, but arising from the collective contribution of all three. Notably, gene expression and protein abundance contributed most strongly to group separation: PD samples were characterized by increased gene expression, while HC samples showed higher proteomic levels. DNA methylation features contributed less to class separation, consistent with their low correlation to the other layers. Finally, functional enrichment analysis of the integrated feature set identified several significantly associated pathways (FDR < 0.05), including PD-relevant biological processes (Fig 3C, S3 Table in S1 File). Similar to the single-omics models, immune and inflammatory responses were implicated across all components. Key processes included cytokine-mediated signaling pathways, such as TNF and IL-1 production and regulation; immune cell recruitment, involving leukocyte and myeloid chemotaxis and natural killer (NK) cell-mediated cytotoxicity; pathogen-host interactions, exemplified by *Yersinia* and *Escherichia coli* infections and antimicrobial defense mechanisms; and receptor-driven activation, highlighted by immune receptor activity (e.g., cytokine and G protein-coupled receptors). Additionally, component-specific enrichments revealed distinct functional modules: Membrane and Vesicle Trafficking: Components 1 and 2 were enriched for terms related to transport/endocytic vesicles and the external side of the plasma membrane, underscoring roles in receptor internalization, signal transduction, and intracellular cargo sorting. Infection and Host Defense: Components 3 and 4 were enriched for pathways linked to bacterial pathogenesis (Component 3) and cytotoxic immune effector functions (Component 4), highlighting coordinated antimicrobial responses. Interestingly, neuroimmune crosstalk was observed in Component 2, exemplified by dopamine receptor binding and its potential interplay with inflammatory signaling, suggesting mechanistic overlaps with PD pathophysiology.

## Network topological analysis

To identify key regulatory genes within the integrated feature space, we applied Hidden Nodes (HN) and Network Propagation (NP) algorithms. These approaches identified topologically central genes that connect or propagate influence among omics features (S4 Table in S1 File). Functional enrichment of these topological regulators—alongside DIABLO-selected features—was performed using the STRING database. The PPI network analysis identified 58 topological regulators and 28 integrated multi-omics features, with 22 of those implicated in significant pathways, such as regulation of MAPK cascade, neuroinflammation and glutamatergic signaling, oxidative stress induced senescence, selective autophagy, activated TAK1 mediates p38 MAPK activation, chaperone mediated autophagy and late endosomal microautophagy. Those pathways span neuroinflammation, glutamatergic signaling, MAPK signaling, autophagy mechanisms, oxidative stress, collectively underscore their potential mechanistic roles in PD pathogenesis (Fig 5). The integration of these pathways highlights the complex interplay between various cellular processes, suggesting that dysregulation in these areas may contribute to PD's pathophysiology.

The results of network stability revealed that the topological regulatory network exhibits small-world properties (average path length = 3.03, clustering coefficient = 0.67) and a high degree of modularity (Q = 0.56), indicating functionally segregated communities with efficient information transfer. The network showed moderate robustness (score = 0.59) and significant centralization (degree centrality = 0.41), with a subset of highly connected hub nodes. Notably, regulatory factors exhibited higher connectivity than their targets (ratio = 1.18), suggesting their central role in coordinating molecular activity. These topological features support the biological relevance and structural stability of the inferred network. Detail results are presented in the S5 Table in S1 File.

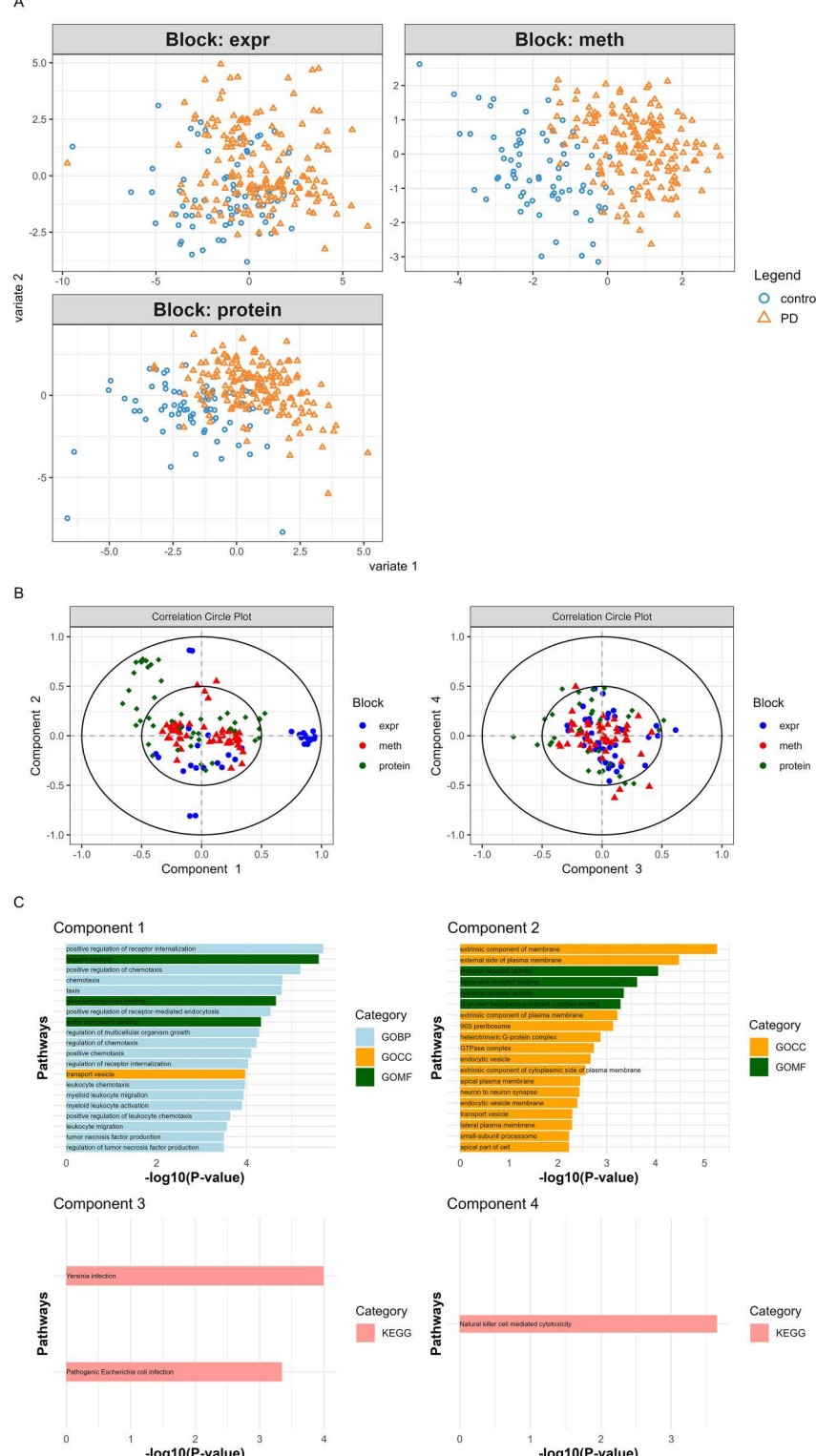

**Fig 3. Multiomics sPLS-DA applied to DNA methylation, transcriptomics, and proteomics data. (A)** Contribution the three Omic datasets into the consensus integrative predictive model. **(B)** Distribution of the selected methylation, genes, and proteins on their corresponding components of multi-omics sPLS-DA. Elements are plotted according to their Pearson's correlation coefficients to the components 1 and 2 ($n = 65$ for DNA methylation, $n = 35$

for genes expression; $n = 45$ for proteins), or 3 and 4 ($n = 60$ for DNA methylation, $n = 45$ for genes; $n = 30$ for proteins). (C) Functional enrichment results on gene set of multi-omics sPLS-DA. Abbreviations: meth = methylation; expr = expression.

## Machine learning

To assess the discriminative capacity of the identified multi-omics signature, along with their key topological regulators derived via systems biology methodologies, we deployed several XGBoost classifiers. The first model, constructed using the multi-omics signature (MO Model), integrated 195 features, comprising 56 methylation CpG sites, 61 genes, and 70 proteins. A subsequent model, based on the topological regulators (TR Model), was formulated employing 59 genes that corresponded to the distinguished features. Detailed information on these features is provided in S6 Table in S1 File. Additionally, a third model, integrated both the multi-omics features and the topological regulators (MO-TR Model). Our finding indicated that the MO Model and MO-TR model achieved the highest AUC of 0.72 and specificity of 0.26, in contrast, the TR model showed the lowest performance for these metrics, with an AUC of 0.62 and a specificity of 0.11. In term of accuracy and F1 scores, the MO Model outperformed the other models, registering values of 0.74 and 0.83, respectively. The accuracy was closely followed by the MO-TR model and the TR model (0.70, 0.69, respectively), while the F1 score was consistent across the MO-TR model and the TR model (0.81). Interestingly, the specificity of all three models was uniformly high, with the MO Model achieving 0.95, the MO-TR model also at 0.95, and the TR model at 0.90.

In the validation dataset, intersection with the test set features yielded 90 features for the MO Model, comprising 28 CpG sites and 62 genes. The TR Model obtained 51 genes, while the MO-TR Model garnered 139 features, including 28 CpG sites and 113 genes. Detail information on these features is showed in S7 Table in S1 File. Performance metrics in the validation set indicated that the TR model exhibited the highest AUC, accuracy, and F1 scores (0.57, 0.62, 0.64, respectively), followed by the MO-TR model (0.56, 0.46, 0.59, respectively) and the MO model (0.54, 0.42, 0.44, respectively). Sensitivity was highest in the MO-TR model (0.83), trailed by the TR and MO models. However, specificity was lowest in the MR-TR model (0.14), succeeded by the MO and TR models (0.36, 0.50, respectively).

Detailed results from the training and testing, and validation datasets are presented in the Tables 3, 4 and Fig 6.

## Discussion

This study investigates the utility of multi-omics integration for the identification of diagnostic biomarkers in PD. Leveraging data from the PPMI cohort, we analyzed a cohort of 305 individuals—comprising 213 PD patients and 92 HCs — with well-balanced demographic characteristics across training and test sets. By integrating DNA methylation and gene expression profiles from peripheral blood with cerebrospinal fluid (CSF)-derived proteomic data, we aimed to capture both peripheral and central pathological signatures while minimizing potential confounders. PD patients were drug-naïve at baseline, which helped eliminate medication-related interference. Analyses of whole blood transcriptomics provide a convenient and less invasive alternative to brain tissue studies for the early diagnosis of PD [35,36]. Given that blood is an easily accessible peripheral biofluid, it serves as a practical substitute due to the substantial transcriptional profile similarities shared between blood and brain tissues. Furthermore, previous research has demonstrated concordance between DNA methylation patterns in peripheral blood and brain tissue, highlighting the clinical utility of these blood-based approaches [25,37]. By minimizing the reliance on invasive brain tissue sampling, our methods offer enhanced practicality for clinical applications.

Our single-omics analysis using sPLS-DA demonstrated that each molecular layer independently possessed discriminative value for distinguishing PD patients from HCs, though with varying efficacy. Among the three omics types, proteomics showed the highest classification accuracy and AUC, followed by transcriptomics, while DNA methylation exhibited the lowest performance. These findings reaffirm that while high-throughput single-omics approaches can uncover disease-associated biomarker [38,39]. For example, a previous study highlighted the potential of piRNAs as

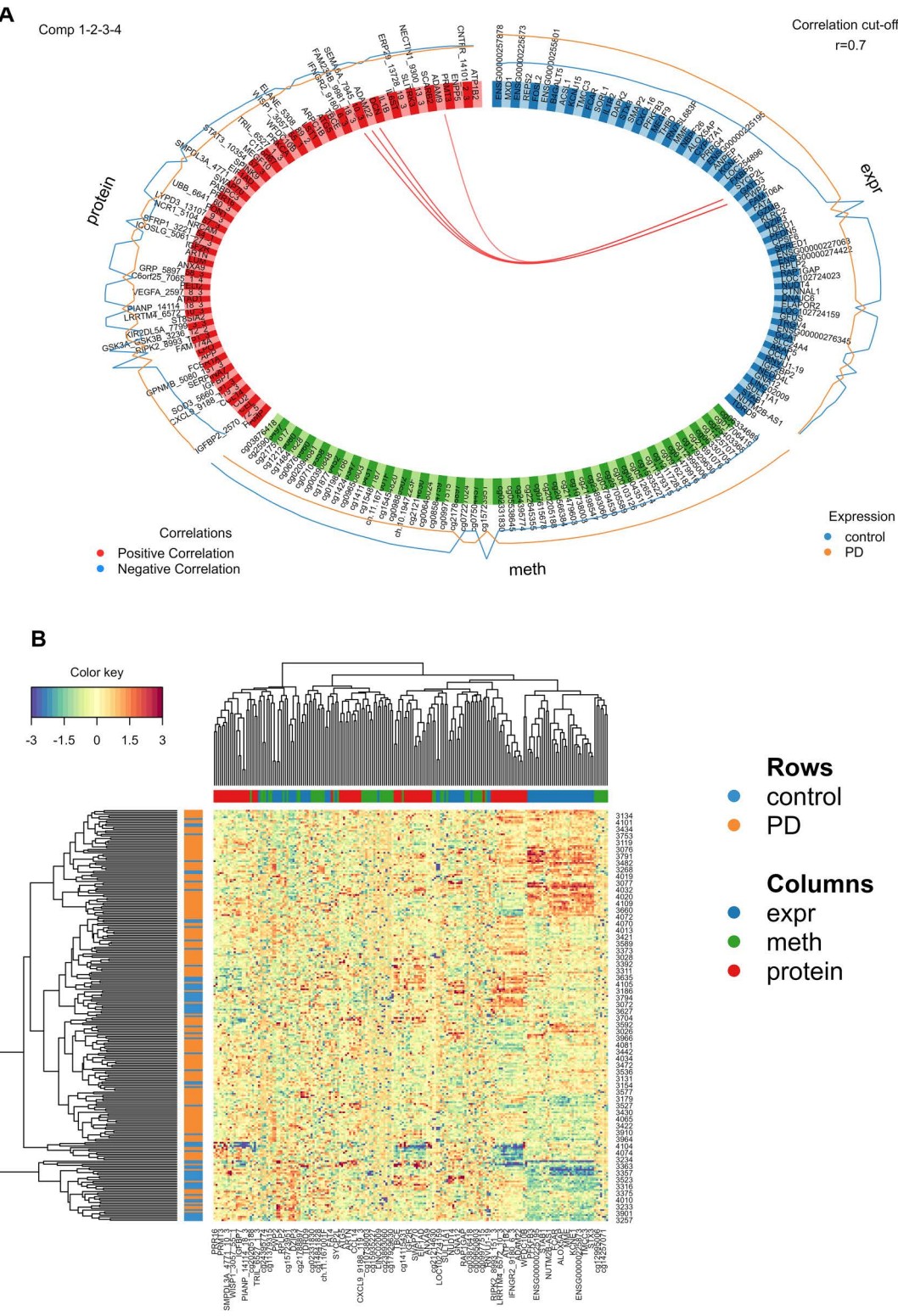

**Fig 4. The relationships among DNA methylation, gene expression, and proteomic profiles across disease groups are characterized through integrative multi-omics visualization. (A)** The Circos plot illustrates the correlations among features derived from three different data. Only variables exhibiting absolute correlation coefficients greater than 0.7 are displayed, emphasizing strong relationships within the dataset. The outer lines represent

the expression levels for each variable across two distinct stage groups, facilitating a comparative analysis of their expression patterns. **(B)** The clustered heatmap presents the variables included in Component 1 from the multi-omics sPLS-DA model. In this visualization, samples are represented in rows, while features from different data types are displayed in columns. Clustering was performed using the Euclidean distance metric and the Complete linkage method. The standardized abundance levels of these features are depicted by the color key.

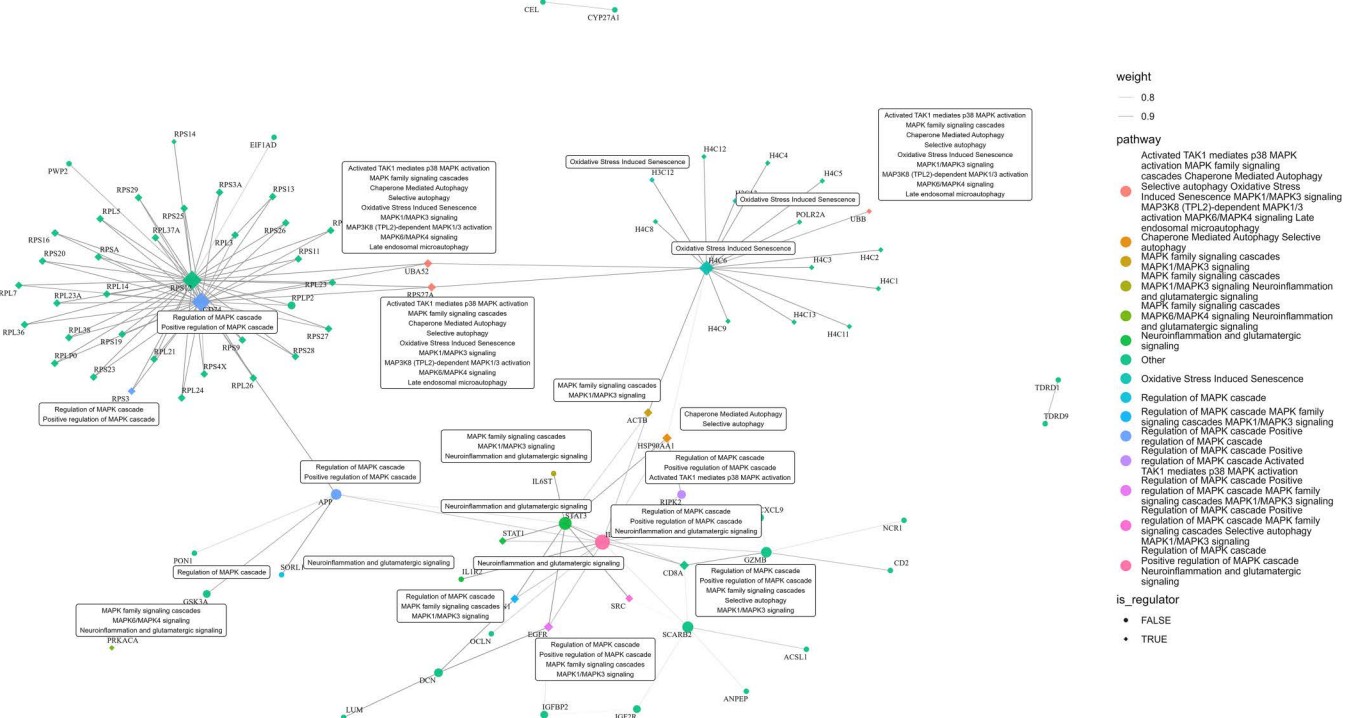

**Fig 5. Integrative PPI network was constructed using genes identified through multi-omics analysis, and their topological regulators.** Nodes include the selected signature genes and regulators with at least one known interaction. The pathways represented within this network provide a detailed view of canonical pathways relevant to the study. These pathways describe various biological processes and signaling cascades.

**Table 3. Test sets result of machine learning models.**

| Models | AUC | accuracy | sensitivity | specificity | F1 score |
|---|---|---|---|---|---|
| MO | 0.72(0.51-0.85) | 0.74(0.61-0.82) | 0.95(0.82-1.00) | 0.26(0.07-0.50) | 0.83(0.73-0.90) |
| TR | 0.62(0.46-0.77) | 0.69(0.54-0.79) | 0.95(0.83-1.00) | 0.11(0-0.34) | 0.81(0.70-0.88) |
| MO-TR | 0.72(0.43-0.84) | 0.70(0.54-0.79) | 0.90(0.76-0.97) | 0.26(0.07-0.50) | 0.81(0.68-0.87) |

Abbreviations: AUC = Area Under the Receiver Operating Characteristic Curve; MO = multi-omics; TF = topological regulators.

**Table 4. Validation sets result of machine learning models.**

| Models | AUC | accuracy | sensitivity | specificity | F1 score |
|---|---|---|---|---|---|
| MO | 0.54(0.39-0.66) | 0.42(0.23-0.58) | 0.50(0.15-0.73) | 0.36(0.14-0.66) | 0.44(0.22-0.67) |
| TR | 0.57(0.37-0.73) | 0.62(0.38-0.77) | 0.75(0.38-0.93) | 0.50(0.18-0.73) | 0.64(0.40-0.82) |
| MO-TR | 0.56(0.38-0.72) | 0.46(0.23-0.62) | 0.83(0.44-1.00) | 0.14(0-0.41) | 0.59(0.34-0.76) |

Abbreviations: AUC = Area Under the Receiver Operating Characteristic Curve; MO = multi-omics; TR = topological regulators.

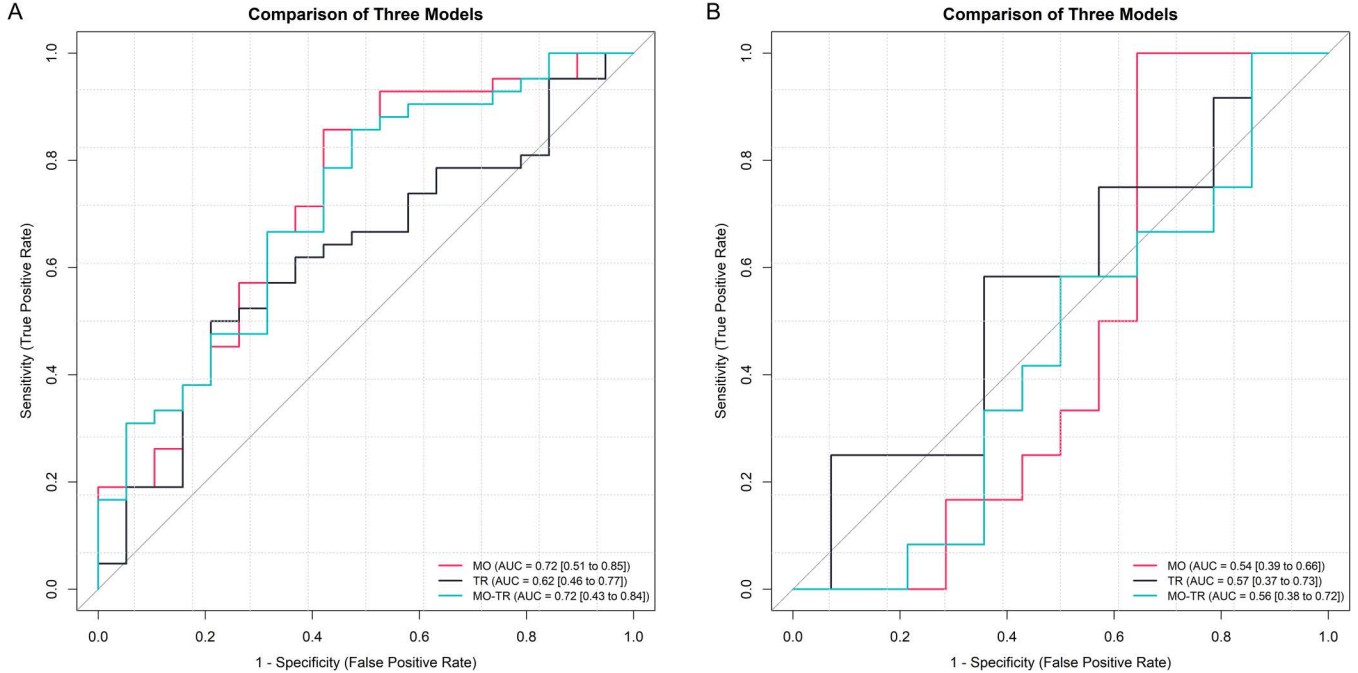

**Fig 6. Model performance on the test and validation datasets. (A)** Receiver operating characteristic curves were utilized to compare the results derived from the test set of three distinct models. (**B**) Receiver operating characteristic curves were utilized to compare the results derived from the validation set of three distinct models. Abbreviations: MO = multi-omics; TR = topological regulators.

effective biomarkers to distinguish PD patients from healthy controls [39]. This underscores the utility of such advanced analytical techniques in identifying robust biomarkers that can aid in the diagnosis and understanding of PD. However, inherent limitations of single omics approaches prevent them from fully capturing the complexity of biological systems, yielding valuable yet incomplete information [40,41]. In particular, proteins are not direct readouts of gene expression due to extensive post-translational modifications such as phosphorylation, glycosylation, and ubiquitination [42], limiting the interpretability of proteomic data in isolation. Thus, relying on any single omics layer may be insufficient for robust diagnostic modeling in PD.

To overcome this limitation, we constructed a DIABLO-based integrative multi-omics sPLS-DA model, which incorporated 56 CpG sites, 61 transcripts, and 70 proteins selected through feature refinement. The integrated model demonstrated improved or comparable performance relative to the best-performing single-omics model, suggesting synergistic benefits. This improved performance likely reflects the optimized feature selection strategy, in which inputs were filtered from features already deemed significant in single-omics modeling, thereby balancing dimensionality reduction with biological relevance. Additionally, although correlations between omics layers were generally weak—especially between DNA methylation and other layers—we observed moderate positive correlations between gene expression and protein abundance. This suggests partial coherence between transcriptional and translational regulation, and highlights the added value of combining complementary information sources in multi-omics modeling. The generally low cross-omics correlations are consistent with the empirical correlation coefficient threshold of 0.1 used in our analysis. While a value of 0.1 may appear weak in absolute terms, it aligns with established observations in multi-omics studies, where biological signals across different molecular layers (e.g., transcriptome, proteome, metabolome) are often attenuated due to biological complexity, technical noise, and post-transcriptional regulation. For instance, Jin et al [32]. reported inter-omics correlation coefficients of approximately 0.1 in integrative analyses of intraplaque hemorrhage-associated cardiovascular signatures.

A correlation threshold of 0.1 reflects a biologically plausible level of connectivity while maintaining model stability and avoiding overfitting, a critical consideration in high-dimensional data settings. Moreover, an excessively strict threshold would yield highly fragmented networks dominated by isolated nodes, thereby limiting the ability to derive integrative biological insights. Therefore, we argue that a threshold of 0.1 represents a conservative and biologically justified estimate that effectively balances sensitivity and specificity in detecting meaningful cross-omics associations.

Interestingly, specific cross-omics associations—such as the correlation between protein expression of *ADAM22* and *ADAM9, SEMA6A*, and the gene expressions of *SYCP1L2* and *GATAD3*—emphasize how multi-omics integration can uncover mechanistic insights obscured in individual layers. These findings underscore that biological complexity often manifests through multi-tiered regulation, which cannot be fully captured by single-modality approaches.

As part of our systems biology exploration, we employed network-based topological analysis using Hidden Node and Network Propagation algorithms to identify key regulatory hubs. Among these, *HSP90AA1* serves as a marker of microglial activation and is associated with neuroinflammation. It participates in the transcriptional programs and pathways linked to the increased fraction of microglia observed in PD [43]. Additionally, *STAT1*, which plays a role in provoking pro-inflammatory microglial activation, has been implicated in of neuroinflammation progress of PD [44]. These findings reinforce the biological interpretability of our integrated model and validate the potential of systems-level modeling for uncovering disease-relevant mechanisms.

A test dataset comprising available multi-omics data and topological regulators was leveraged as a validation dataset to evaluate the performance of the identified features as a diagnostic signature. One of the strengths and confirmations of our integrative multi-omics approach is its ability to identify several known PD candidate genes, including *DNAJC6*, *SORL1*, and *NEDD4L*, which have been reported in previous studies [45–49]. Among the known genes, Mutations in *DNAJC6* cause juvenile-onset, atypical parkinsonism and early-onset PD [46,50]. *DNAJC6* encodes for auxilin 1, which plays a critical role in synaptic vesicle endocytosis, Studies in animal models have shown that auxilin deficiency results in impaired synaptic vesicle endocytosis, thereby negatively impacting synaptic neurotransmission, homeostasis, and signaling [51]. Furthermore, *DNAJC6* mutations are associated with key pathological features of PD, including midbrain-type dopamine neuron degeneration, abnormal α-synuclein aggregation, increased intrinsic neuronal firing frequency, and mitochondrial and lysosomal dysfunctions observed in human midbrain-like organoids [46]. *SORL1* is responsible for encoding *SORLA*, a sorting receptor that interacts with the retromer complex [52]. The genetic association of *SORL1* with Alzheimer's disease has been well established [53]. However, recent findings have revealed that mutations in *SORL1* or its role as a modifier gene were identified in a family presenting features of PD [48]. Additionally, polymorphisms within the *SORL1* gene may contribute to the susceptibility to PD in the northern Chinese population [47]. Further studies are required to investigate the potential inherent correlation between the *SORL1* gene and the pathogenesis of PD. *Nedd4L*, also known as *Nedd4−2*, is a member of the HECT (homologous to E6-associated protein C-terminus) family of E3 ubiquitin ligases. It is known for its role in mediating the ubiquitination of various proteins, including the epithelial sodium channel (ENaC), voltage-gated sodium channels (Navs), and glutamate transporters [54,55]. Previous research has demonstrated that *Nedd4−2* mediates the ubiquitination of glutamate transporters both in vitro and in vivo using PD models. This study indicates that the knockdown of *Nedd4−2* results in improvements in motor deficits and tyrosine hydroxylase (TH) expression in PD mice by enhancing the levels of glutamate transporters. These findings suggest that *Nedd4−2* may be considered a potential therapeutic target for the treatment of PD [56].

In our validation dataset, we were only able to acquire partial DNA Methylation and Expression Profiles of Whole Blood. The CpG sites of Methylation in the validation set (GEO database) are half of those in the training set (PPMI cohort). This is due to the fact that whole-genome methylation profiling in PPMI was conducted using Illumina Human Methylation EPIC, while the methylation examination in GEO was performed using Illumina Infinium HumanMethylation450K Beadchips. After interacting with training set, we obtained partial genes in our validation dataset. Unfortunately, we were unable to obtain the proteinic data necessary to validate the performance of the training set. As a result, we lost the advantage of

integrating different levels of biological information. The absence of key features and their interactions—critical for model performance—likely contributed to suboptimal the results on the validation set. Additionally, differences in platform and cohort characteristics, as well as clinical heterogeneity between two datasets, such as variations in medication use and diagnostic criteria, may have further compromised the model's generalizability and validation performance.

The low specificity in ML results was observed in the test and validation sets. Several factors may contribute to this finding. First, class imbalance in the data: The PPMI cohort used for training contains a relatively small number of healthy controls compared to PD patients. This imbalance can bias the classifier toward predicting PD, leading to high sensitivity but reduced specificity. Second, limited feature specificity: Although many of the selected features are implicated in PD-related pathways, some may also be involved in broader neurodegenerative or inflammatory processes (e.g., *SORL1*). This overlap may reduce their discriminative power in distinguishing PD from normal aging or other neurological conditions, thereby affecting specificity.

Although our integrative multi-omics approach holds considerable promise, it also presents several limitations. Firstly, this study is based exclusively on data from the PPMI cohort and the GEO database, with analyses inherently limited by the availability and scope of these public resources. While our findings are robust within this context, we acknowledge the importance of experimental validation to confirm our results. Unfortunately, wet-lab follow-up experiments are currently beyond the scope of this work due to limitations in access to matched biospecimens, laboratory infrastructure, and funding. As a secondary analysis of publicly available omics datasets, our study lays a computational foundation that we hope will guide future experimental investigations. Secondly, our multi-omics approach was limited to DNA methylation, gene expression, and protein data. Incorporating metabolomic and microbiomic data into the multi-omics framework could potentially enhance the robustness and predictive power of our model, and offer a more comprehensive understanding of the underlying biological mechanisms. Thirdly, the PPMI cohort and the GEO dataset (GSE165083) are predominantly composed of individuals of European ancestry, recruited from specialized academic centers in North America and Europe. Genetic and epigenetic biomarkers can exhibit population-specific variability due to differences in allele frequencies, methylation patterns, gene expression regulation, and environmental exposures associated with ancestral and geographic backgrounds. To date, most publicly available Parkinson's disease omics datasets, including those in the GEO database, share similar demographic limitations. This restricts the generalizability of our findings and limited our ability to conduct comprehensive cross-ethnic validation. Future studies in more diverse populations will be essential to ensure the broader applicability of identified biomarkers.

## Conclusion

In conclusion, our study provides strong evidence supporting the utility of integrative multi-omics modeling for early and accurate diagnosis of PD. Through the combination of robust statistical learning, topological network inference, and machine learning validation, we identify both novel and known disease-associated features that deepen our understanding of PD biology and may inform future biomarker development and therapeutic targeting.

## Supporting information

**S1 File. S1 Table.** Bootstrap Stability Scores of Selected Features in sPLS-DA. **S2 Table.** Lists of the selected features in single- and multi-omics analysis. **S3 Table.** Results of function enrichment. **S4 Table.** The identified topological regulator. **S5 Table.** Table Stability Assessment of Network Topological analysis. **S6 Table.** Table Features of the train and test sets in machine learning. **S7 Table.** Table Features of the validation set in machine learning. **S1 Fig.** Balanced error rates (BER) of training models using sPLS-DA applied to DNA methylation, gene expression and proteomics datasets. A range of features from 3 to 300 (from 3 to 30 in steps of 3, from 30 to 60 in steps of 6, from 60 to 150 in the step of 15, and from 150 to 300 in the step of 30) was set for each component. Results were calculated with stratified 5-fold cross-validation and 100 random repeats. **S2 Fig**. (A)The performance of prediction models was evaluated under optimal parameter

settings, with results measured by accuracy and AUC using stratified 5-fold cross-validation and 1,000 random repeats. The findings are presented as means ± standard deviation (SD). The notations MTEH, EXPR, PROT represent the outcomes of individual sPLS-DA analyses based on expression, methylation, and proteins respectively. The term MULTI denotes the results of an integrative sPLS-DA analysis across the three omics. (**B**) An overlapped features among three omics data across four distinct components identified through multi-omics integrative analysis. METH, methylation; EXPR, expression; PRO, proteins; MULTI, multi-omics.
(ZIP)

## Author contributions

**Conceptualization:** Wei Liu, Xuejing Wang, Jiuqi Wang.

**Data curation:** Wei Liu.

**Formal analysis:** Wei Liu, Xuejing Wang, Jiuqi Wang.

**Investigation:** Wei Liu, Xuejing Wang, Jiuqi Wang.

**Methodology:** Wei Liu, Xuejing Wang, Jiuqi Wang.

**Project administration:** Xuejing Wang, Jiuqi Wang.

**Resources:** Wei Liu, Xuejing Wang, Jiuqi Wang.

**Software:** Wei Liu.

**Supervision:** Lina Xu, Xuejing Wang, Jiuqi Wang.

**Validation:** Wei Liu, Xuejing Wang, Jiuqi Wang.

**Visualization:** Wei Liu, Xuejing Wang, Jiuqi Wang.

**Writing – original draft:** Wei Liu.

**Writing – review & editing:** Xuejing Wang, Jiuqi Wang.

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
