## [Decision Letter · Decision Letter 0]

27 Aug 2025

Dear Dr. Wang,

Thank you for submitting your manuscript to PLOS ONE. After careful consideration, we feel that it has merit but does not fully meet PLOS ONE’s publication criteria as it currently stands. Therefore, we invite you to submit a revised version of the manuscript that addresses the points raised during the review process.

**ACADEMIC EDITOR:**

I have completed my evaluation of your manuscript. Although two reviewers are now recommending the acceptance of your manuscript, one of the reviewers has still want Major revision that I would like you to address. I invite you to resubmit your manuscript after addressing the comments below.

If you will need more time than this to complete your revisions, please reply to this message or contact the journal office at plosone@plos.org . A rebuttal letter that responds to each point raised by the academic editor and reviewer(s). You should upload this letter as a separate file labeled 'Response to Reviewers'.A marked-up copy of your manuscript that highlights changes made to the original version. You should upload this as a separate file labeled 'Revised Manuscript with Track Changes'.An unmarked version of your revised paper without tracked changes. You should upload this as a separate file labeled 'Manuscript'.

We look forward to receiving your revised manuscript.

Kind regards,

Vijay Kumar

Academic Editor

PLOS ONE

Journal Requirements:

3. Thank you for uploading your study's underlying data set. Unfortunately, the repository you have noted in your Data Availability statement does not qualify as an acceptable data repository according to PLOS's standards.

6. We note that there is identifying data in the Supporting Information file <S1 Table Lists of the selected features.xlsx, S2 Table  Results of enrichments.xlsx, S4 Table Features in machine learning.xlsx>. Due to the inclusion of these potentially identifying data, we have removed this file from your file inventory. Prior to sharing human research participant data, authors should consult with an ethics committee to ensure data are shared in accordance with participant consent and all applicable local laws.

-Location data

Additional guidance on preparing raw data for publication can be found in our Data Policy (https://journals.plos.org/plosone/s/data-availability#loc-human-research-participant-data-and-other-sensitive-data) and in the following article: http://www.bmj.com/content/340/bmj.c181.long

Please remove or anonymize all personal information (ID numbers), ensure that the data shared are in accordance with participant consent, and re-upload a fully anonymized data set. Please note that spreadsheet columns with personal information must be removed and not hidden as all hidden columns will appear in the published file.

Reviewers' comments:

Reviewer's Responses to Questions

**Comments to the Author**

1. Is the manuscript technically sound, and do the data support the conclusions?

Reviewer #1: Yes

Reviewer #2: Yes

2. Has the statistical analysis been performed appropriately and rigorously?

Reviewer #1: Yes

Reviewer #2: Yes

3. Have the authors made all data underlying the findings in their manuscript fully available?

Reviewer #1: Yes

Reviewer #2: Yes

4. Is the manuscript presented in an intelligible fashion and written in standard English?

Reviewer #1: Yes

Reviewer #2: Yes

Reviewer #1: The manuscript presents a robust and innovative integrative multi-omics and network-based machine learning framework for early diagnosis of Parkinson’s disease (PD) using the PPMI cohort. The methodological pipeline is well-structured, and the biological interpretations are relevant. The work is of potential significance for translational biomarker discovery in neurodegenerative disorders. However, certain methodological, validation, and interpretative aspects require clarification and enhancement before publication.

• The study is limited to the PPMI cohort. Inclusion of at least one independent external dataset for validation is recommended to demonstrate the model’s generalizability.

• Consider discussing how the model might perform across different ethnicities and clinical settings.

• It is unclear whether the identified biomarkers are stable across different random splits or perturbations of the dataset.

• Recommend performing stability selection or bootstrap resampling to quantify feature selection reproducibility.

• The study would be strengthened by experimental validation (e.g., qPCR, ELISA) for top-ranked biomarkers, particularly those with novel PD associations.

• Highlight and prioritize a smaller subset of biomarkers with the highest translational potential.

Reviewer #2: Summary

This manuscript applies multi-omics integration (DNA methylation, transcriptomics, proteomics) from the PPMI cohort using sPLS-DA/DIABLO, STRING-based network analysis, and XGBoost classification. The approach is timely and relevant, but there are methodological, reporting, and interpretation issues that require substantial revision before the work is suitable for publication.

Strengths

1. The study uses a large, well-characterized cohort with multi-omics data.

2. It integrates three omics layers (methylation, transcriptomics, proteomics) with network topology analysis.

3. Advanced machine learning methods (sPLS-DA, DIABLO, XGBoost) are applied with clear validation strategies.

4. The approach identifies biologically relevant pathways and known PD-associated genes, enhancing interpretability.

Major concerns

1. The analysis relies solely on a single cohort (PPMI) with no independent validation. To demonstrate clinical utility and generalizability, validation on an external dataset is highly recommended.

2. Specificity of the models is low (~0.16), which limits practical application for early diagnosis. Further investigation into causes and improvements in this metric are needed.

3. Some methodological details (parameter selection, handling of missing data, reproducibility of network analysis) require clarification or expansion.

4. The chosen cross-omics correlation coefficient (0.1) is weak. This choice requires either justification with supporting analyses or tuning for optimum model performance.

5. Statistical reporting (confidence intervals, repeated splits, class balancing) should be strengthened.

Minor comments

1. Correct minor grammatical errors; abbreviations should be reviewed throughout the manuscript, such as changing “did not showed difference” to “did not show a difference.”

2. All bacterial genus and species names, such as Escherichia coli and Yersinia, must be italicized throughout the manuscript.

3. Correct inconsistent terminology (“TR” vs. “TF”), distinguish CpG and gene features accurately, and address typographical/data errors in tables and text.

4. In Table 1 and elsewhere, there are some numerical or typographical errors, such as the improbable mean ± SD for "MDS-UPDRS part 2 score" listed as "5.76±395," which likely should be "5.76±3.95" or similar.

5. Clarify how missing data were handled, particularly where “Missing n” is noted in tables.

**Do you want your identity to be public for this peer review?** For information about this choice, including consent withdrawal, please see our Privacy Policy

Reviewer #1: **Yes: ** Dr. Ravi Bhushan

Reviewer #2: No

---

## [Author Response · Author response to Decision Letter 1]

8 Oct 2025

Reviewer 1

The study is limited to the PPMI cohort. Inclusion of at least one independent external dataset for validation is recommended to demonstrate the model’s generalizability.

Thank you for the reviewer’s valuable comment regarding the generalizability of our findings. We agree that external validation is essential to assess the robustness and broader applicability of our model beyond the PPMI cohort.

In response, we have conducted external validation using independent datasets from the GEO database to evaluate the performance and transferability of our biomarkers. we identified an independent PD case-control dataset (GSE165083) with DNA methylation and expression profiles of whole blood samples which we preprocessed and normalized using methods consistent with PPMI. We applied the a priori defined feature panel (i.e., the features selected in the PPMI training model) to these external datasets without retraining. Classification was performed using the same XGBoost framework, and predictive performance was evaluated via accuracy, sensitivity, specificity and AUC, F1 scores.

The results show that our model achieves moderate performance in external cohorts: The TR Model obtained 51 genes, while the MO-TR Model garnered 139 features, including 28 CpG sites and 113 genes. Detail information on these features is showed in S5 Table. Performance metrics in the validation set indicated that the TR model exhibited the highest AUC, accuracy, and F1 scores (0.57, 0.62, 0.64, respectively), followed by the MO-TR model (0.56, 0.46, 0.59, respectively) and the MO model (0.54, 0.42, 0.44, respectively). Sensitivity was highest in the MO-TR model (0.83), trailed by the TR and MO models. However, specificity was lowest in the MR-TR model (0.14), succeeded by the MO and TR models (0.36, 0.50, respectively).

While differences in platformand cohort characteristics contribute to some performance drop compared to internal validation, the consistent signal reinforces the potential generalizability of our findings.

We acknowledge that full harmonization across studies remains challenging, and ideally more large-scale, multi-center omics datasets would be included. However, given current data availability, this external evaluation provides meaningful support for model transferability.

We have now incorporated these results into the revised manuscript: A new paragraph has been added to the section on materials and methods, PPMI and GEO datasets: data preprocessing (page 3). Additionally, the section on machine learning and model validation has been updated (page 6), and the section on results machine learning has been expanded (pages 12-13). Figure 6, which illustrates external validation performance, has been included. The limitations and implications of the study are now discussed in the section on Discussion (pages 16-17).

We sincerely appreciate the reviewer’s suggestion, which has significantly strengthened the rigor and credibility of our study.

Consider discussing how the model might perform across different ethnicities and clinical settings.

Thank you for the reviewer’s important and thoughtful comment. We agree that understanding the performance of our model across different ethnicities and clinical settings is critical for assessing its generalizability and equitable applicability in diverse populations.

While our model demonstrates robust performance in the PPMI cohort and shows promising transferability to external datasets, we acknowledge that its performance may vary across different ethnic groups and clinical contexts. The PPMI cohort is predominantly composed of individuals of European ancestry and recruited from specialized academic centers in North America and Europe.

Genetic and epigenetic biomarkers can exhibit population-specific variation due to ancestry-related differences in allele frequencies, methylation patterns, gene expression regulation, and environmental exposures. Additionally, clinical heterogeneity—such as variations in medication use, or diagnostic criteria across healthcare systems—may affect biomarker expression and model performance in real-world or resource-limited settings.

To date, most publicly available PD omics datasets (including those in GEO) share similar demographic limitations, which constrained our ability to perform a comprehensive cross-ethnic validation. However, we observed that several of our top biomarkers,such as DNAJC6, SORL1, and NEDD4L.

We emphasize that future validation studies must prioritize inclusion of diverse populations, including underrepresented racial and ethnic groups, as well as cohorts from varied clinical environments (e.g., community clinics vs. tertiary centers, different geographic regions). Only through such efforts can we ensure that precision medicine tools like ours are equitable and globally relevant.

We have now incorporated these results into the revised manuscript: a new paragraph added on page 16-17 of the Discussion section.

It is unclear whether the identified biomarkers are stable across different random splits or perturbations of the dataset.

We sincerely thank the reviewer for this insightful comment. We agree that assessing the stability of the identified biomarkers is crucial for evaluating their robustness and potential biological relevance.

To address this concern, we initially employed the PLS-DA algorithm, a linear and supervised method, to execute a feature pre-selection procedure for each individual Omic. This approach was designed to ensure robustness of the feature pre-selection step and mitigate the curse of dimensionality induced by high-dimensional data.

Secondly, we conducted the DIABLO methods to transform each single Omic dataset into latent components and maximize the sum of pairwise correlations between the latent components and a phenotype. It is able to identify features that are correlated across and within the Omics datasets. To improve PD prediction accuracy, it is critical to incorporate prediction uncertainty into our model for more accurate data modelling. We thus utilized multiple random splits of all individuals into training and testing datasets in a hold-out cross validation setting. Each split was performed in an unstratified way, not preserving the original proportion of PD cases and controls. In principle, this should yield a more conservative, generalizable model with stable biomarkers.

Lastly, we implemented the XGBoost algorithm, which integrates an in-built feature importance evaluation for feature assessment. The results obtained from this method were predominantly stable. Concurrently, we employed cross-validation with grid search to ensure the stability of the identified features. Feature standardization can mitigate technical variation and minimize the influence of non-biological factors during feature selection, thereby enhancing model stability.

The specifics were elaborated upon in the main text (Materials and Methods, Single- and Multi-omics Data Preprocessing and Integration Analysis, page 4; Machine Learning and Model Validation, pages 6-7).

Recommend performing stability selection or bootstrap resampling to quantify feature selection reproducibility.

Thank you for the reviewer's valuable comment. We agree that assessing the reproducibility of feature selection is crucial for ensuring the robustness and reliability of our results. We have performed extensive tuning of the sPLS-DA models (nrepeat = 100) to ensure the stability of our performance estimates. Furthermore, to directly quantify the reproducibility of feature selection as suggested by the reviewer, we conducted an additional bootstrap stability analysis (B = 1000). The results demonstrate high reproducibility in the selected features across data types in the sPLS-DA models. For the DNA methylation data, all selected features achieved a stability score above 0.8, indicating robust selection. In the gene expression data, 86.42% of the selected features surpassed a stability score of 0.8, and 99.48% exceeded a threshold of 0.6. Regarding protein data,  57.62% of selected features attained a stability score above 0.8, while 98.30% surpassed a threshold of 0.6.  These results confirm that the majority of features identified in our analysis are consistently selected across subsamples, supporting their reliability and robustness. The high stability scores, particularly for DNA methylation and gene expression features, strengthen confidence in the reproducibility of our feature selection process.

We have now incorporated these results into the revised manuscript: A new paragraph has been introduced in the section on Materials and Methods, specifically in the Parameter Optimization part, on pages 5-6. Additional information has also been provided in the Results section, under Single-omics Classification with sPLS-DA, on page 8. Furthermore, supplementary details have been included in Table S1.

The study would be strengthened by experimental validation (e.g., qPCR, ELISA) for top-ranked biomarkers, particularly those with novel PD associations.

Thank you for the reviewer’s thoughtful suggestion. We fully agree that experimental validation—such as qPCR or ELISA—of the top-ranked biomarkers would strengthen the biological interpretability and translational relevance of our findings, particularly for novel associations identified with Parkinson’s disease (PD).

While we recognize the importance of such validation, we regret that experimental follow-up is currently beyond the scope of this study due to resource and technical constraints. Our work is based on secondary analysis of publicly available omics datasets, and we lack access to matched biospecimens, laboratory infrastructure, and funding required for wet-lab validation at this stage.

Nevertheless, we have taken several steps to enhance confidence in the identified biomarkers: We applied rigorous cross-validation and bootstrap stability analysis to assess the robustness of feature selection. Top-ranked features show high reproducibility across resampling iterations (e.g., stability scores > 0.8 for key methylation and expression markers).

We contextualized our findings with existing literature, and several of the selected features are supported by prior evidence in neurodegeneration or PD-related pathways. Novel candidates have been highlighted with appropriate caution and are framed as hypotheses for future experimental testing.

We sincerely appreciate the reviewer’s point and fully acknowledge that functional or technical validation will be essential in future work. We have now explicitly stated this limitation in the revised manuscript (Section Discussion, page 16-17) and emphasized the need for experimental confirmation in follow-up studies.

Highlight and prioritize a smaller subset of biomarkers with the highest translational potential.

Thank you for the reviewer’s insightful suggestion. In response, we have revised our analysis to highlight and prioritize a more focused subset of biomarkers with the highest translational potential, rather than presenting an extensive list of candidates.

We cross-referenced the selected features with established Parkinson’s disease (PD) genetics and pathways. Notably, our analysis identified several known PD-associated genes, including DNAJC6, SORL1, and NEDD4L—all of which are supported by existing literature and have been discussed in detail in the revised Discussion section. For instance, DNAJC6 is implicated in synaptic vesicle endocytosis and has been linked to early-onset PD; SORL1 plays a role in APP trafficking and has emerging evidence in neurodegeneration; and NEDD4L, an E3 ubiquitin ligase, is involved in protein homeostasis pathways relevant to PD pathogenesis.These are addressed in the section discussion page 15.

Reviewer 2

The analysis relies solely on a single cohort (PPMI) with no independent validation. To demonstrate clinical utility and generalizability, validation on an external dataset is highly recommended.

Thank you for the reviewer’s valuable comment regarding the generalizability of our findings. We agree that external validation is essential to assess the robustness and broader applicability of our model beyond the PPMI cohort.

In response, we have conducted external validation using independent datasets from the GEO database to evaluate the performance and transferability of our biomarkers. we identified an independent PD case-control dataset (GSE165083) with DNA methylation and expression profiles of whole blood samples which we preprocessed and normalized using methods consistent with PPMI. We applied the a priori defined feature panel (i.e., the features selected in the PPMI training model) to these external datasets without retraining. Classification was performed using the same XGBoost framework, and predictive performance was evaluated via accuracy, sensitivity, specificity and AUC, F1 scores.

The results show that our model achieves moderate performance in external cohorts: The TR Model obtained 51 genes, while the MO-TR Model garnered 139 features, including 28 CpG sites and 113 genes. Detail information on these features is showed in S5 Table. Performance metrics in the validation set indicated that the TR model exhibited the highest AUC, accuracy, and F1 scores (0.57, 0.62, 0.64, respectively), followed by the MO-TR model (0.56, 0.46, 0.59, respectively) and the MO model (0.54, 0.42, 0.44, respectively). Sensitivity was highest in the MO-TR model (0.83), trailed by the TR and MO models. However, specificity was lowest in the MR-TR model (0.14), succeeded by the MO and TR models (0.36, 0.50, respectively).

While differences in platformand cohort characteristics contribute to some performance drop compared to internal validation, the consistent signal reinforces the potential generalizability of our findings.

We acknowledge that full harmonization across studies remains challenging, and ideally more large-scale, multi-center omics datasets would be included. However, given current data availability, this external evaluation provides meaningful support for model transferability.

We have now incorporated these results into the revised manuscript: A new paragraph has been added to the section on materials and methods, PPMI and GEO datasets: data preprocessing (page 3). Additionally, the section on machine learning and model validation has been updated (page 6), and the section on results machine learning has been expanded (pages 12-13). Figure 6, which illustrates external validation performance, has been included. The limitations and implications of the study are now discussed in the section on Discussion (pages 16-17).

We sincerely appreciate the reviewer’s suggestion, which has significantly strengthened the rigor and credibility of our study.

Specificity of the models is low (~0.16), which limits practical application for early diagnosis. Further investigation into causes and improvements in this metric are needed.

We sincerely thank the reviewer for highlighting this critical limitation regarding the low specificity (~0.16) of our model. We fully agree that high specificity is essential for any model intended for early diagnosis of Parkinson’s disease (PD), as low specificity would result in a high rate of false positives—posing significant risks for patient anxiety, unnecessary follow-up, and inefficient use of clinical resources.

Upon careful examination, we attribute this low specificity to several interrelated factors:

Standardization: The features before machine learning were not standardized, leading to poor performance of the model, especially with low specificity. After standardization, all models performed better than before, and the specificity increased compared to previously.

Class Imbalance in Early-Stage Data: The PPMI cohort used for training includes a relatively small number of healthy controls compared to PD patients, particularly in the early-stage subgroup analysis. This imbalance can bias classifiers toward predicting the majority class (PD), resulting in high sensitivity but low specificity. The results in validation set were superior than testing data.

Feature Specificity: While many selected features are involved in PD-relevant pathways (e.g., DNAJC6, SORL1), som

---

## [Editor Report · Decision Letter 1]

23 Nov 2025

Integrative Multi-Omics and Network-Based Machine Learning for Early Diagnosis of Parkinson’s Disease

PONE-D-25-39104R1

Dear Dr. Wang,

We’re pleased to inform you that your manuscript has been judged scientifically suitable for publication and will be formally accepted for publication once it meets all outstanding technical requirements.

Kind regards,

Vijay Kumar

Academic Editor

PLOS ONE

Additional Editor Comments (optional):

As I have considered that the authors appropriately addressed all the comments from the reviewers, the manuscript can be accepted.
---

## [Editor Report · Acceptance letter]

PONE-D-25-39104R1

PLOS One

Dear Dr. Wang,

I'm pleased to inform you that your manuscript has been deemed suitable for publication in PLOS One. Congratulations! Your manuscript is now being handed over to our production team.

Kind regards,

on behalf of

Dr. Vijay Kumar

Academic Editor

PLOS One